

# A revised turtle assemblage from the Upper Cretaceous Menefee Formation (New Mexico, North America) with evolutionary and paleobiostratigraphic implications

Brent Adrian[1], Heather F. Smith[1,2] and Andrew T. McDonald[3]

[1] School of Human Evolution and Social Change, Arizona State University, Tempe, Arizona, United States
[2] Department of Anatomy, Midwestern University, Glendale, Arizona, United States
[3] Western Science Center, Hemet, California, United States

Corresponding author
Brent Adrian, badrian@asu.edu

## ABSTRACT

The middle Campanian Menefee Formation (Fm.) of the San Juan Basin of New Mexico is a relatively understudied terrestrial deposit in southern Laramidia preceding the fossil-rich upper Campanian Fruitland and Kirtland formations that have been studied for more than a century. Previous collection efforts have revealed a diverse dinosaurian and crocodyliform fauna in the Menefee Fm., including ankylosaurian, tyrannosaurid, hadrosaurid, ceratopsian, and neosuchian taxa. Nearly a decade has passed since the last description of the Menefee turtle fauna, and we present new material that provides insight into turtle evolution during the Campanian in the Colorado Plateau, with implications for Late Cretaceous turtle paleobiostratigraphy in Laramidia. In particular, we extend the stratigraphic ranges of the baenids *Neurankylus baueri* and *Scabremys ornata* backwards from younger San Juan Basin strata, along with the nanhsiungchelyid *Basilemys*. Additional material increases Menefee representation of the relict helochelydrid *Naomichelys*, and the regionally prevalent derived baenids *Denazinemys* and *Thescelus*. Additionally, we report new pan-trionychian specimens, which provide insight into the persistence of *Adocus* and multiple trionychid and plastomenid species through the remainder of the Campanian in the San Juan Basin. A cluster analysis of turtle diversity across early-middle Campanian sites in Laramidia shows distributions consistent with latitudinal provinciality in some groups. For instance, derived baenids were restricted to latitudes south of southern Utah, along with marine taxa (bothremydids and protostegids) and pan-kinosternoids. Basin-scale endemism is also suggested by some baenid and trionychid distributions. Otherwise, the turtle fauna of the Menefee most closely resemble those of the similarly-aged Wahweap Fm. in southern Utah, and the Aguja Fm. in the Big Bend area of Texas and Mexico to a lesser extent. The Menefee turtle assemblage is consistent with reconstructed paleoenvironments characteristic of the western shoreline of the Western Interior Seaway. Recurrent cyclothems in these settings shaped the development of fluviodeltaic deposits that preserved distal components of large channels with surrounding floodplains and swamps, along with evidence of freshwater, brackish, and possibly shallow marine influence.

# INTRODUCTION

Since 2011, yearly expeditions have been conducted to the Upper Cretaceous Menefee Formation (Fm.) in the San Juan Basin (SJB) of northwestern New Mexico (*Beaumont, Dante & Sears, 1956*; *Miller, Carey & Thompson-Rizer, 1991*) (Fig. 1). These efforts have involved researchers, students, and volunteers from various institutions, including the Western Science Center, Southwest Paleontological Society, and Zuni Dinosaur Institute for Geosciences. To this point, this fieldwork has resulted in the descriptions of the tyrannosaurid *Dynamoterror dynastes McDonald, Wolfe & Dooley, 2018*, the ankylosaurian *Invictarx zephyri McDonald & Wolfe, 2018*, and the brachylophosaurin hadrosaur *Ornatops incantatus McDonald et al., 2021*. *Menefeeceratops sealeyi Dalman et al. (2021)* has also been described from the Menefee Fm. and could be the oldest member of the Centrosaurinae. Remains of the giant alligatoroid *Deinosuchus Holland, 1909* represent one of the earliest occurrences in Laramidia and North America more broadly, demonstrating a stratigraphic range extension in the SJB (*Mohler, McDonald & Wolfe, 2021*).

Initial reports of turtles from the Menefee Fm. were mostly of fragmentary material from indeterminate baenids and trionychids across 13 localities (*Hunt & Lucas, 1993*). More recently, *Lichtig & Lucas (2015)* provided brief descriptions and two figures showing five Menefee turtle specimens. Remains are attributed to adocid (aff. *Adocus bossi Gilmore, 1919*), baenid (*Denazinemys Lucas & Sullivan, 2006* and indeterminate), trionychid taxa, a single specimen of the helochelydrid *Naomichelys Hay, 1908*, and a partial putative bothremydid shell (CHCU 81269). Perhaps the most comprehensive documentation of Upper Cretaceous turtles from the San Juan Basin was provided by *Sullivan, Jasinski & Lucas (2013)*, focusing on the late Campanian (Kirtlandian) Fruitland and Kirtland formations. Additional studies focusing on these younger units include *Sullivan et al. (1988)*, *Lucas & Sullivan (2006)*, and *Jasinski et al. (2018)*. The current study updates the taxonomic composition of the Menefee turtle assemblage, discusses stratigraphic range extensions for several taxa, and makes comparisons with stratigraphically correlative units to discuss paleobiogeographical patterns during the early to middle Campanian in Laramidia.

## Geological setting

The Menefee Formation is part of the Mesaverde Group and is underlain by the regressive Point Lookout Sandstone and overlain by the transgressive Cliff House Sandstone (*Collier, 1919*; *Sears, 1925*; *Sears, Hunt & Dane, 1936*; *Sears, Hunt & Hendricks, 1941*; *Pike, 1947*; *Beaumont, Dante & Sears, 1956*; *Molenaar, 1983*) (Fig. 1). The Menefee Fm. is divided into two formal members: the basal Cleary Coal Member (*Beaumont, Dante & Sears, 1956*) and the overlying Allison Member (originally the "Allison barren member" (*Sears, 1925*; *Sears, Hunt & Dane, 1936*)). The nomenclature of the uppermost coal-bearing portion of the

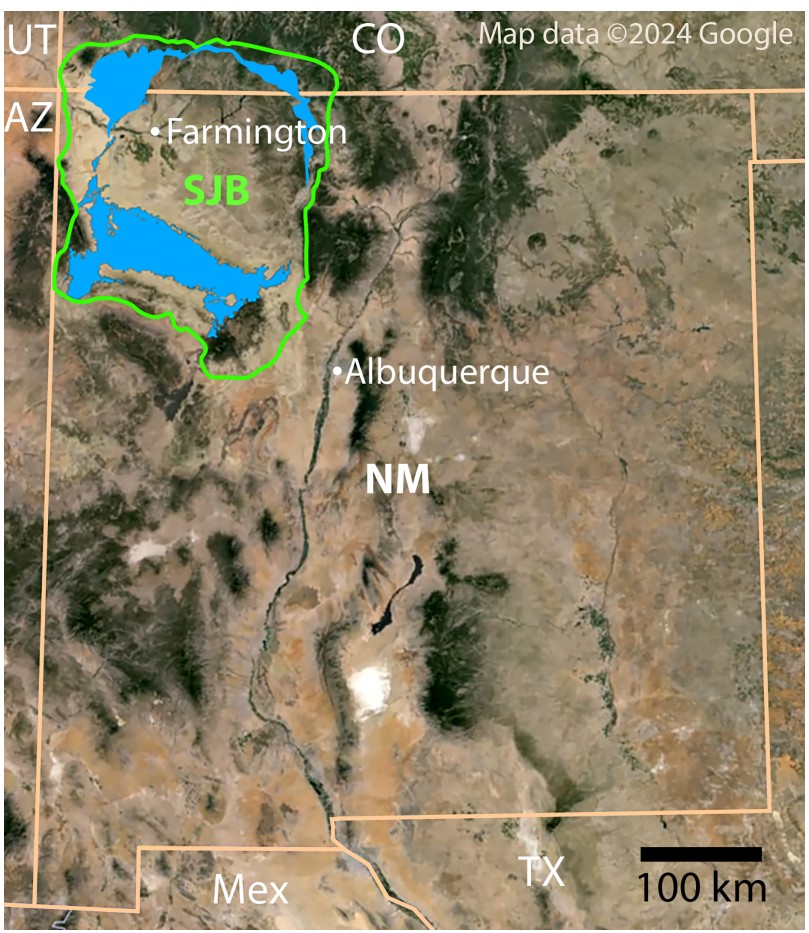

**Figure 1 Index map of New Mexico and surrounding areas.** Green outline indicates boundary of San Juan Basin and the Menefee Formation is shaded blue (*Collier, 1919*; *Haynes, Vogel & Wyant, 1972*; *Scholle, 2003*). Figure created using Google Earth Pro (Image Landsat/Copernicus) and Adobe Illustrator v.28.4.1. Source map © 2023 Colorado Plateau Geosystems, Inc. Scale = 100 km.

Menefee Fm. has been contentious, with some authors recognizing an unnamed upper coal member (*e.g.*, *Hayes & Zapp, 1955*; *Beaumont & Hoffman, 1992*). Alternatively, *Sears, Hunt & Dane (1936)* considered the uppermost coal-bearing strata to be part of the Allison Member. *Miller (1984)* and *Miller, Carey & Thompson-Rizer (1991)* agreed and proposed three informal subdivisions of the Allison Member, in ascending order: the Lower Beds, characterized by the lack of coal beds and calcareous concretions; the Juans Lake Beds, characterized by the presence of calcareous concretions; and the La Vida Beds, characterized by the absence of calcareous concretions and the presence of thin coal beds. The presence of calcareous concretions and lack of coal in the beds that produced the turtle fossils described here indicate that they pertain to the Juans Lake Beds (*Miller, 1984*; *Miller, Carey & Thompson-Rizer, 1991*). For further detailed description of the local geology of the study area, see *Mohler, McDonald & Wolfe (2021)*.

The geochronological age of the Menefee Formation at different locations across its vast outcrop belt is difficult to determine due to the diachronous nature of its strata

(*Peterson & Kirk, 1977*: fig. 2) and lack of an extensive sample of radioisotopic dates (*Lucas et al., 2005*). An age of 78.22 ± 0.26 Ma was derived from a bentonite layer near the top of the Menefee Fm. in the Gallina hogback in the northeastern San Juan Basin (*Lucas et al., 2005*). The ammonoid index fossil *Baculites perplexus* has been reported from the Cliff House Sandstone in Chaco Canyon in the vicinity of and on depositional strike from our study area (*Siemers & King, 1974*). In this area, the uppermost Menefee Fm. intertongues with and is overlain by the upper tongue (= Chacra Mesa tongue (*Sears, Hunt & Dane, 1936*; *Beaumont & Hoffman, 1992*)) of the Cliff House Sandstone (*Donselaar, 1989*). *Molenaar et al. (2002)* showed an approximate age of between 78.5 Ma and 78.0 Ma for *B. perplexus*. Therefore, the turtles, crocodylomorphs, dinosaurs, and other fossils from our study area could be somewhat older than 78.0 Ma, but radioisotopic dating is necessary for greater accuracy and precision.

## MATERIALS AND METHODS

Fossil specimens were collected under permits NM11-005S, NM12-03S, NM16-11S, NM18-03S, and NM24-04S issued by the U.S. Bureau of Land Management (BLM) and are reposited at the Western Science Center in Hemet, California; Natural History Museum of Utah in Salt Lake City, Utah; and New Mexico Museum of Natural History & Science in Albuquerque, New Mexico. The new turtle material described here was collected in the study area in San Juan County, New Mexico; exact locality data are on file at the BLM and the respective repositories. The National Park Service and Chaco Culture National Historical Park gave access and permission to study CHCU 81269 under permit CHCU-2023-SCI-0008. We apply the taxonomic scheme of turtles presented by *Joyce (2007*, *2017*), and adhere to Phylocode guidelines unless otherwise indicated (*e.g.*, *Laurin et al., 2005*; *Joyce et al., 2021*). Following *Zangerl (1969)*, the two pairs of scales present on the anterior plastron are termed gular and intergular scales, where the intergulars are located anterior to the gular scales, and both sets of scales are anterior to the entoplastron.

To recognize patterns in the paleobiogeographic distribution of lower to middle Campanian turtle taxa along the eastern coast of Laramidia, we compiled faunal lists of turtle assemblages from geological units stratigraphically correlated with the Menefee Fm. using the regional stratigraphic correlations of *Beveridge et al. (2022*: fig. 9). Additional turtle assemblages were compared for the Aguja and Mesaverde formations in the Big Bend region of northern Mexico and Texas and northwestern Wyoming, respectively (see references in *López-Conde, Chavarría-Arellano & Montellano-Ballesteros (2020)*, *Wu et al. (2023)*). A hierarchical cluster analysis (paired group UPGMA using a Jaccard similarity index) was performed on the binary matrix of lower to middle Campanian turtle assemblages using PAST software (*Hammer, Harper & Ryan, 2001*). Particular trionychid and plastomenid species were not included due to uncertainty of taxonomic consistency across the sampled sites. The Jaccard similarity index was selected because of its capabilities with binary data and strong performance in Mesozoic biogeographic studies (*e.g.*, *Schmachtenberg, 2008*).

# RESULTS

## Systematic paleontology

Testudinata *Klein, 1760*

Baenidae *Cope, 1873*

*Denazinemys Lucas & Sullivan, 2006*

*Denazinemys nodosa* (*Gilmore, 1916*)

Fig. 2

Referred specimens: WSC 10769, posterior plastral lobe; WSC 10770, carapace fragment; NMMNH P-97386, right costal 1 fragment.

**Description.** WSC 10769 is the most intact specimen attributable to *Denazinemys nodosa* that we have collected, representing most of the posterior plastral lobe (Figs. 2A, 2B). It was broken unevenly from the rest of the plastron, and the lack of visible sutures in the specimen suggests the individual was developmentally mature (Fig. 2A). The right xiphiplastron is missing several pieces, especially posteriorly. A prominent ridge runs along the free margin on the ventral side, positioned almost a centimeter (cm) from the edge (Fig. 2A). The ridge likely represents the site of integumentary attachment and the body wall boundary. The shape of WSC 10769 is triangular and narrower posteriorly with a shallow anal notch that is slightly more developed than in the *D. nodosa* holotype (USNM 8345) (*Sullivan, Jasinski & Lucas, 2013*: fig. 20.2b). The preservation of any ventral sulcus pattern was disrupted by damage to the external cortex surrounding the midline, and the ventral surface is otherwise smooth and free of ornament (Fig. 2B). The typical extensive fusion known for baenids also obscures the pattern of sutures defining the posterior extent of the hypoplastron and the shape of the articulation with the xiphiplastra (*Joyce & Lyson, 2015*). However, short segments of the femoral-anal sulci are intermittenly intact in WSC 10769 and incised deeply, measuring almost a millimeter across. These segments suggest that the femoral-anal sulcus is straight rather than omega-shaped, which is the diagnostic shape for the sulcus in *Denazinemys* and also the derived group of baenids, Baenodda (*Brinkman, 2003*; *Joyce & Lyson, 2015*; *Spicher et al., 2023*). This would seem to undermine our referral of WSC 10769 to *Denazinemys*, except that the holotype USNM 8345 from the late Campanian Kirtland Fm. also has a nearly straight femoral-anal sulcus (*Sullivan, Jasinski & Lucas, 2013*: fig. 20.2b). This suggests that the shape of the sulcus is variable in *D. nodosa*, which should be reflected in future revisions of its diagnosis. The extent of variation and any temporal or geographic patterns for this trait are unknown due to low sample sizes, but could be investigated further as more material of this taxon is recovered.

The posterior lobe is wider at the level of the contact between the femoral-anal sulcus and the shell margins for WSC 10769 than for the holotype (Table 1). Similar to the holotype, the margins of the xiphiplastra at this contact are slightly indented in WSC 10769 (Fig. 2B). The lateral ends of the posterior plastral lobe occur near the contact between the abdominal-femoral sulcus and the shell margin at the base of the lobe, which

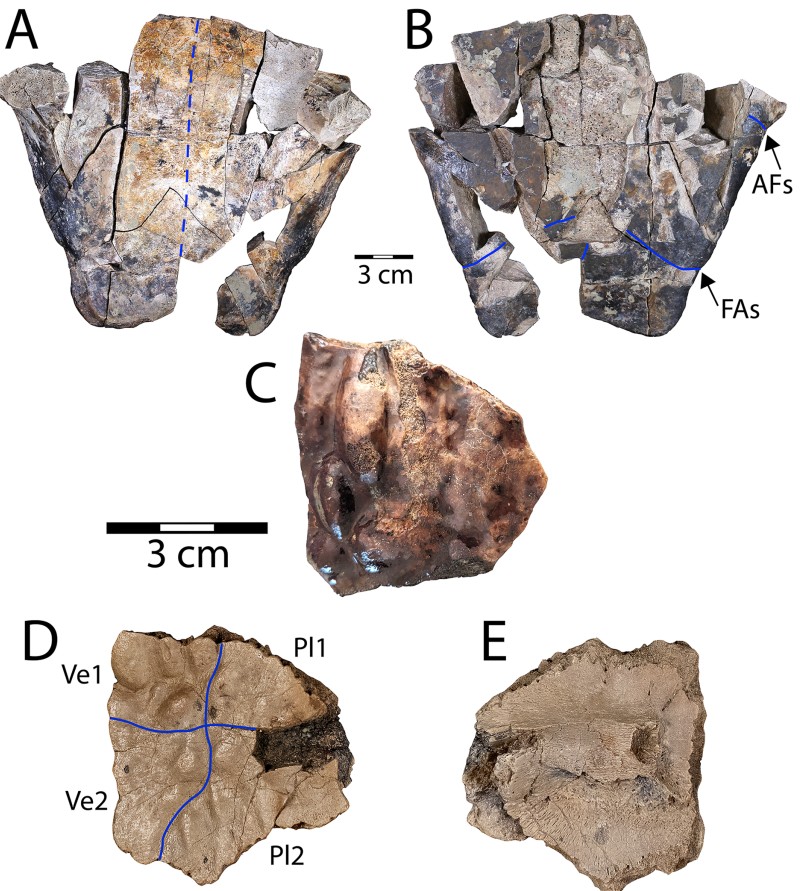

**Figure 2** *Denazinemys nodosa* **posterior plastral lobe (WSC 10769).** (A) Dorsal, and (B) ventral views; (C) dorsal surface of carapace fragment (WSC 10770) showing characteristic welt-like sculpture; (D) dorsal and (E) ventral sides of costal 1 (NMMNH P-97386). Note different scale for C–E (each 3 cm); solid blue lines indicate sulci and dashed blue line shows location of fused sutures. Abbreviations: AFs = abdominal-femoral sulcus; FAs = femoral-anal sulcus; Pl = pleural scale; Ve = vertebral scale.

is slightly wider in WSC 10769 compared to the holotype (Table 1; Fig. 2B). However, the length of the posterior plastral lobe, measured from the left terminus of the abdominal-femoral sulcus is almost the same between the two specimens.

WSC 10770 is an isolated carapacial fragment with a well-preserved example of the pronounced rounded, lumpy dorsal carapace texture of *D. nodosa* (Fig. 2C). The protrusions are distinguished from those of the nodosely-sculptured *Scabremys* (*Sullivan, Jasinski & Lucas, 2013*) by their generally rounded, ovoid, welt-like character. In contrast, the protrusions in *Scabremys* more closely resemble parasagittal rectilinear ridges that are more regular, antero-posteriorly oriented, and longer proportionally than in *Denazinemys* (*Lucas & Sullivan, 2006*; *Sullivan, Jasinski & Lucas, 2013*).

NMMNH P-97386 is a right costal 1 from a subadult *D. nodosa*, which is apparent due to lack of fusion with surrounding bony elements (*Hutchison, 1984*; *Joyce & Lyson, 2015*)

**Table 1 Dimensions of the posterior plastral lobe and sulci (in cm).** Measurements from the *Denazinemys nodosa* holotype USNM 8345, WSC 10769, a line drawing of the *Scabremys ornata* holotype USNM 13229 by *Gilmore (1935)* (reproduced by *Sullivan, Jasinski & Lucas, 2013*: fig. 20.3b-c), and UMNH VP 28352.

| Specimen | Abdominal-femoral sulcus width | Femoral-anal sulcus width | Posterior plastral lobe length | Straight carapace length | Straight plastral length |
|---|---|---|---|---|---|
| *Denazinemys nodosa* holotype (USNM 8345) | 14.3 | 10.1 | 10.5 | 41.5 | 38.0 |
| WSC 10769 (*Denazinemys* sp. posterior plastral lobe) | 18.2 | 12.1 | 10.6 | — | — |
| *Scabremys ornata* holotype (USNM 13229) line drawing (*Gilmore, 1935*) | 15.4 | 10.5 | 11.5 | 42.5 | 38.6 |
| UMNH VP 28352, nearly complete articulated *Scabremys* sp. shell | 16.0 | 10.7 | 9.9 | 43.0 | 39.0 |

(Figs. 2D, 2E). The dorsal surface is covered with the characteristic nodose texture of *D. nodosa*, and pustules vary in size and shape. Sulci are shallow and extend between pustules forming a cruciform pattern that divides the dorsal surface between the first two vertebral scales medially and the first two pleural scales on the lateral side (Fig. 2D). On the ventral side, the first thoracic rib is wide and low, crossing the surface transversely and terminating at an ovoid rib head (see *Joyce, Schoch & Lyson, 2013*) (Fig. 2E).

*Neurankylus Lambe, 1902*

*Neurankylus baueri Gilmore, 1916*

Fig. 3

Referred specimens: WSC 10612, partial articulated shell; CHCU 81269, partial plastron

**Description.** WSC 10612 is a partial articulated shell that includes approximately the anterior half of the carapace and most of the plastron. The anterior margin of the carapace is convex anteriorly, and the anterior marginal edge is rounded. WSC 10612 can be referred to *Neurankylus baueri* based on its large size and the arrangement of sulci on the ventral surface of the anterior plastral lobe (Figs. 3C, 3D). In particular, the gular scales are well-developed without meeting at the midline, and the intergular scales are U-shaped, contacting the humeral scales posteriorly (*Larson et al., 2013*; *Sullivan, Jasinski & Lucas, 2013*; *Joyce & Lyson, 2015*; *Lichtig & Lucas, 2018*). The bones of the shell are entirely fused and there are no discernable sutures, as is typical for baenids. The co-ossification of the shell, in addition to pervasive fracturing, limits the ability to discern some morphologies. However, the combination of bone and scale morphologies described above is consistent with those known for *N. baueri* (see summary of *Neurankylus* in *Joyce & Lyson (2015)*). Preserved sulci typically have rough edges, reaching 2.4 millimeters (mm) wide and ~1 mm deep. They define the extent of marginal scales 1–2, vertebral scale 1, the medial side of marginal scale 3, the vertebral-pleural sulci on the left side, the right lateral edge and posterolateral corner of the cervical scale, and the right pleural-marginal sulcus, including the posterolateral corner of pleural scale 1 (on costal 2) (Figs. 3A, 3B). Costals 1–4 are preserved on the left side, along with costals 1–2 on the right. The cervical scale appears to be single, undivided, rectangular, and wider than long. The vertebral scales are also wider than long. Vertebral scale 1 contacts the cervical scale, marginal scales 1 and 2, and pleural

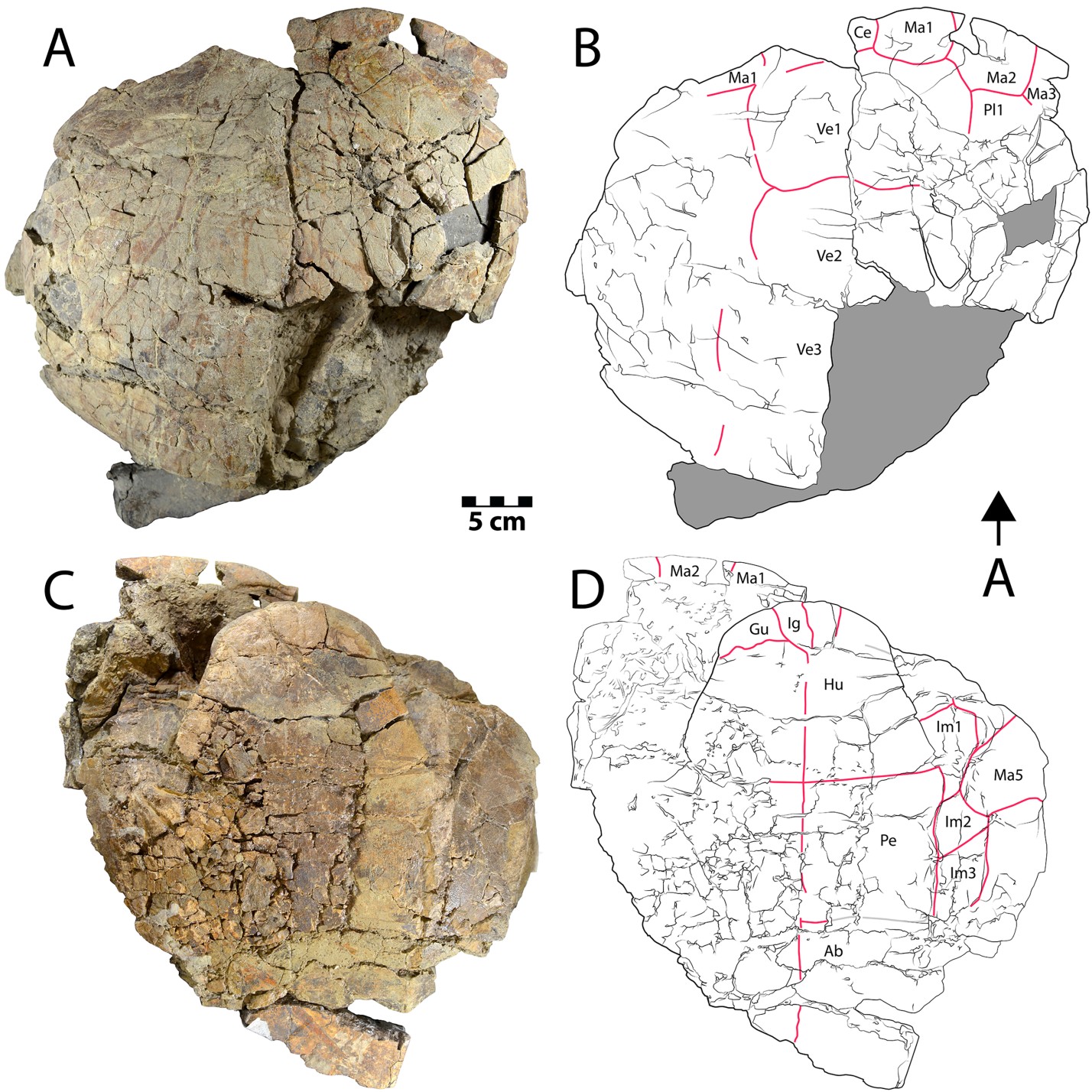

**Figure 3 Shell of *Neurankylus baueri* (WSC 10612).** Dorsal (A) photograph and (B) line drawing; ventral view (C) photograph and (D) line drawing. Abbreviations: Ab = abdominal scale, Ce = cervical scale, Gu = gular scale, Hu = humeral scale, Ig = intergular scale, Im = inframarginal scale, Ma = marginal scale, Pe = pectoral scale, Pl = pleural scale, Ve = vertebral scale. Blue lines indicate sulci; arrow points in anterior ("A") direction; scale = 5 cm.

scale 1. Marginal scale 2 contacts marginal scales 1 and 3, pleural scale 1, and vertebral scale 1. The surfaces of the carapace and plastron are nearly smooth, consistent with previous descriptions (*e.g.*, *Larson et al., 2013*; *Joyce & Lyson, 2015*). Table S1 provides additional measurements for some discernable bones and scales.

Similar to the carapace, the plastron is heavily fractured and missing most of the posterior lobe (Figs. 3C, 3D). The anterior plastral lobe is wider than long and rounded anteriorly. In addition to the arrangement of gular/intergular scales described above, most of the humeral, pectoral, and femoral scales are at least partially preserved, along with left inframarginal scales 1–3, and partial marginals along the ventral bridge on the left side. Based on comparison with the reconstruction of *N. baueri* by *Gilmore (1916)*, the length of the carapace of WSC 10612 is estimated to be 584 mm, exceeding the diagnostic threshold of 500 mm for *Neurankylus* spp. (*Joyce & Lyson, 2015*). The number of comparable linear measurements for WSC 10612 was limited due to complete shell fusion and extensive fracturing, and vertebral scale proportions were not available for SMP VP-2379 due to the fractured condition of the carapace. However, the two specimens from the Kirtland Fm. (USNM 8344 and SMP VP-2379) are otherwise proportionally similar, consistent with comparisons by *Sullivan, Jasinski & Lucas (2013)*, especially in midline lengths of the plastral scales (Fig. 4). The greatest difference between the three specimens involves the size of the anterior plastral lobe, which is smallest in both length and width in the Menfee specimen WSC 10612, despite estimation of the specimen to be largest in total maximum carapace length. Proportional differences could be due to dimorphism, but probably not ontogeny due to overall size similarity. The specimens from the Kirtland Fm. are also several million years younger so morphological differences could reflect changes in plastral dimensions over time (*Lichtig & Lucas, 2018*).

CHCU 81269 is a partial plastron of a turtle that was recovered from Chaco Culture National Historical Park in 2007. *Santucci et al. (2015)* briefly mentioned the specimen as a pelomedusid pleurodiran (side-necked turtle), while *Lichtig & Lucas (2015)* provided a morphological assessment and referred it to the bothremydid taxon *Elochelys* cf. *E. perfecta* based primarily on purported similarities in the posterior plastral lobe sulci. The ventral surface of the posterior plastral lobe is the only portion of CHCU 81269 with a discernable pattern of sulci (Figs. 5A, 5C). The posterior plastral lobe is a fused, thick plate of bone that is missing its posterior rim (Fig. 5C). Shell fusion and lack of sutures are unknown among the numerous pleurodiran taxa (including bothremydids) surveyed by *Gaffney, Tong & Meylan (2006)*, though it is a key trait of baenids (*Hutchison, 1984*; *Joyce & Lyson, 2015*). Additional fusion is also apparent in a previously unidentified fragment (*Lichtig & Lucas, 2015*: fig. 9D), which is identified here as a sequence of fused neurals, also consistent with the pattern observed in Baenidae (see examples in *Smith et al. (2017)*). Critically, the dorsal surface of the posterior plastral lobe also lacks any evidence of articular scars associated with pelvic (ischial and pubic) articulations, which are commonly-recognized synapomorphies for Pleurodira that would have been evident and diagnostic for bothremydids (*Gaffney, Tong & Meylan, 2006*; *Mayerl et al., 2017*: fig. 1) (Fig. 5B). The pattern of sulci on the posterior plastral lobe of CHCU 81269 is generally similar to the published pholidoses of *Elochelys perfecta* (*Nopsca, 1931*; *Gaffney, Tong & Meylan, 2006*:

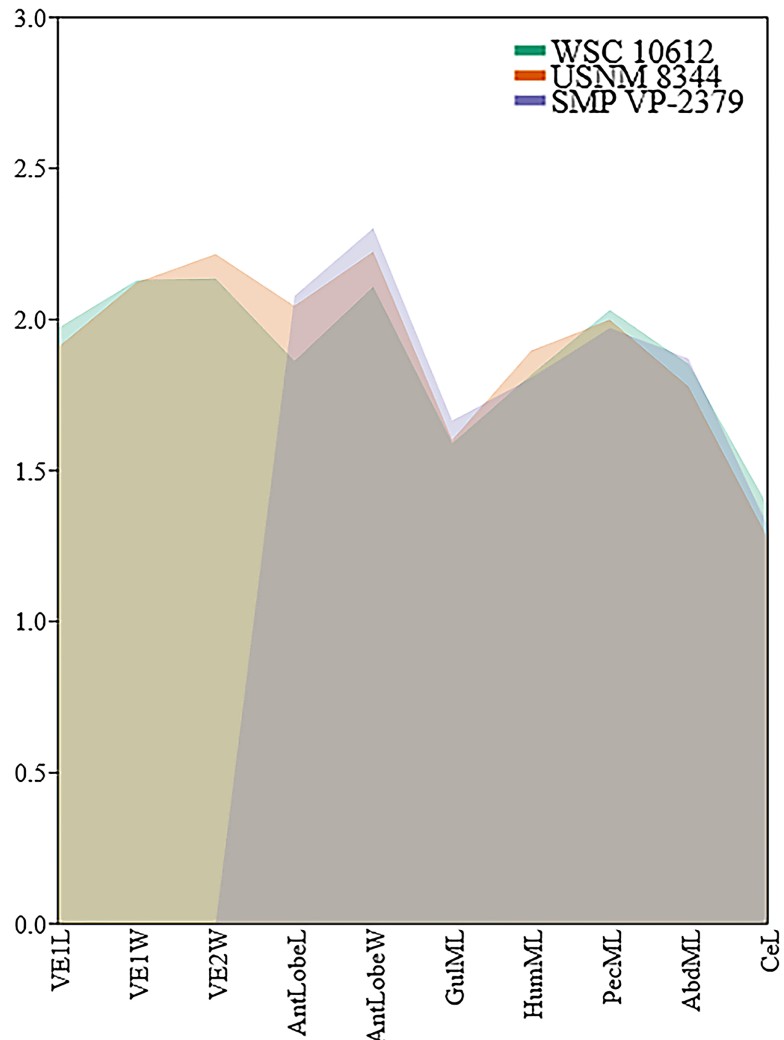

**Figure 4 Comparison of log-transformed linear shell measurements among *Neurankylus baueri* specimen WSC 10612, SMP VP-2379 (*Sullivan, Jasinski & Lucas, 2013*: fig 20.6c, d), and the holotype USNM 8344 using measurements from the text of *Gilmore (1916)*.** Abbreviations: Abd = abdominal scale, AntlobeL = anterior plastral lobe length, AntlobeW = anterior plastral lobe width, Ce = cervical scale, Gul = gular scale, Hum = humeral scale, L = maximum length; ML = midline length; Pec = Pectoral scale; VE = vertebral scale, W = maximum width.

fig. 270C–D). However, it is also similar to that of *Neurankylus baueri* in both sulcus arrangement and posterior plastral lobe shape (*Sullivan, Jasinski & Lucas, 2013*: fig. 20.6a–b) (Fig. 5C), which are well documented in specimens from the San Juan Basin (*Lichtig & Lucas, 2016*, *2018*). CHCU 81269 also lacks a prominent anal notch and we were unable to confirm a small mesoplastron such as those of bothremydids (*Gaffney, Tong & Meylan, 2006*; *Pérez-García et al., 2017*). Also, the locations of sulci in the bridge region were consistent with the anterior inframarginal scales of baenids (*Gaffney, Tong & Meylan, 2006*; *Joyce & Lyson, 2015*). The length and width of the posterior plastral lobe (11 and 14.5 cm, respectively) of CHCU 81269 are somewhat smaller than the *N. baueri* holotype

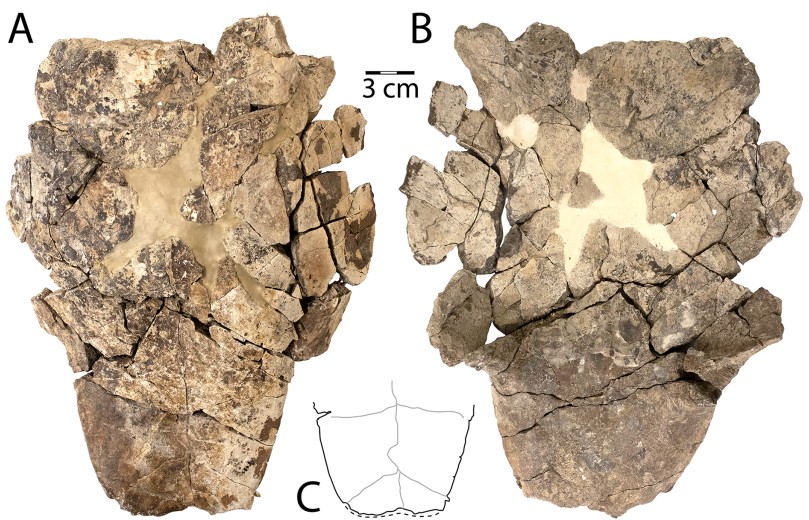

**Figure 5 CHCU 81269, a partial plastron of *Neuranklyus baueri*, previously attributed to the bothremydid *Elochelys* cf. *E. perfecta* (*Lichtig & Lucas, 2015*).** (A) Ventral view, (B) dorsal view, (C) pholidosis of posterior plastral lobe, with sulci indicated in gray. Dotted line indicates reconstructed edge. Scale = 3 cm.

(USNM 8344), but larger than the reconstructed plastron of *E. perfecta* (*Gaffney, Tong & Meylan, 2006*; *Sullivan, Jasinski & Lucas, 2013*). The lack of shell texture noted for CHCU 81269 is also more consistent with *N. baueri* than for pleurodires, which typically have a variant of a characteristic "pelomedusoid" pattern (see *Gaffney, Tong & Meylan, 2006*). In summary, the traits that favor an identification of *Neurankylus baueri* for CHCU 81269 rather than *Elochelys* cf. *E. perfecta* include: extensive shell fusion and a lack of discerable sutures; absent pelvic articular structures on the dorsal surface of the xiphiplastron; missing small, laterally located mesoplastra; bridge sulci consistent with baenid inframarginal scale pattern; anal notch lacking; and the absence of a typical vermiform pelomedusoid shell texture. Based on these characteristics and the lack of parsimony incumbent with referral to an extra-continental bothremydid taxon, we refer CHCU 81269 to *N. baueri*.

*Scabremys Sullivan, Jasinski & Lucas, 2013*
*Scabremys ornata* (*Gilmore, 1935*)
Fig. 6
Referred specimen: UMNH VP 28352, nearly complete articulated shell.

**Description.** UMNH VP 28352 is a partial, articulated shell with a complete plastron and a partial carapace that is preserved in its entire length (Fig. 6). As in the *N. baueri* specimen WSC 10612 described above, the shell is filled in by dark gray mudstone and a dark brown sideritic concretion, entirely co-ossified with no discernable sutures, and is badly fractured. The shell is referrable to *Scabremys ornata* and differs from the coeval and similar *Denazinemys nodosa* (see *Lucas & Sullivan, 2006*). Unlike *D. nodosa*, the carapace of UMNH VP 28352 is oval, with the widest point midway along the carapace

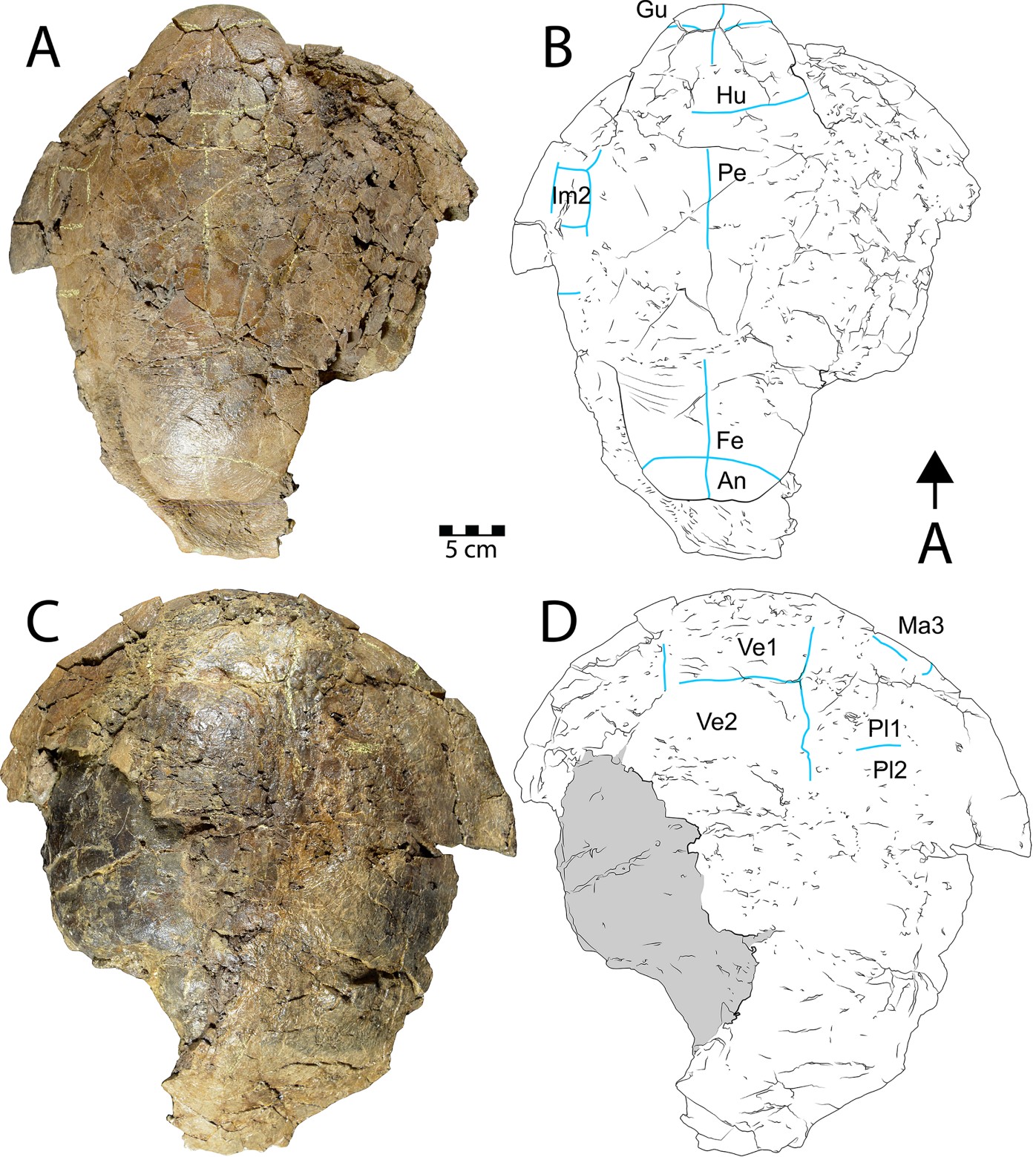

**Figure 6 Shell of *Scabremys* sp. (UMNH VP 28352).** Ventral (A) photograph and (B) line drawing; dorsal view (C) photograph and (D) line drawing. Abbreviations: An = anal scale, Gu = gular scale, Fe = femoral scale, Hu = humeral scale, Im = inframarginal scale, Ma = marginal scale, Pe = pectoral scale, Pl = pleural scale, Ve = vertebral scale. Blue lines indicate sulci; arrow points in anterior ("A") direction; scale = 5 cm.

(*Sullivan, Jasinski & Lucas, 2013*: fig. 20.3). The shell sculpture of *S. ornata* is nodose and similar to *D. nodosa*, but is composed of raised, longitudinal ridges what are somewhat regular in length and alignment along the entire carapacial midline, with narrow grooves between them (*Sullivan, Jasinski & Lucas, 2013*). The sculptural morphology of UMNH VP 28352 is consistent with *S. ornata* (Figs. 6C, 6D). In comparison, the sculpture of *D. nodosa* is more rounded, irregular, and not ridge-like (see below). The ridges of *S. ornata* are concentrated on the first four vertebral scales and the medial sides of the costals, but are more laterally extensive in *D. nodosa* (*Sullivan, Jasinski & Lucas, 2013*: figs. 20.3a, 20.2a). Unlike *D. nodosa*, the first vertebral scale of UMNH VP 28352 is hexagonal and inferred to be widest anteriorly (*Sullivan, Jasinski & Lucas, 2013*; *Joyce & Lyson, 2015*). Intergular scales are absent and the gular scales are not divided, which is the diagnostic arrangement for *S. ornata* (*Sullivan, Jasinski & Lucas, 2013*) (Figs. 6A, 6B). A few intermittent sulci are preserved on the carapace of UMNH VP 28352, indicating the posterior and lateral edges of vertebral scale 1, the right lateral edge of vertebral scale 2, the contact between pleural scales 1 and 2 on the right side, as well as edges of an anterior (cf. third) marginal scale on the right side (Fig. 6D).

The posterior edge of the carapace and part of the right margin are preserved and have rounded edges in cross section. The posteriormost edge of the carapace is flat and transitions on the right side to a postero-laterally facing, slightly concave segment of the margin (Figs. 6C, 6D). Unlike *Denazinemys*, the posterior edge of the carapace extends beyond the plastron (*Sullivan, Jasinski & Lucas, 2013*). Vertebral scale 5 reaches the rounded posterior carapace margin in both species (*Lucas & Sullivan, 2006*; *Sullivan, Jasinski & Lucas, 2013*). The distinctive ridged surface texture of *S. ornata* is preserved on the right anterior side of the dorsal carapace and near the midline (Fig. 6C). Otherwise, the shell surface is nearly smooth, with texture formed by short, fine wrinkles.

Sulci on the ventral side of the anterior plastral lobe of UMNH VP 28352 define the contact between the gular, humeral, and pectoral scales (Figs. 6A, 6B). The gular-humeral and humeral-pectoral sulci are mostly straight and approximately perpendicular to the midline. The anterior plastral lobe is wider than long, broadly convex anteriorly, and extends slightly past the anteriormost margin of the carapace. The posterior plastral lobe is approximately the same size as the anterior lobe and is also rounded, but it has a flat posteriormost margin with no anal notch (Figs. 6A, 6B). The shape of the posterior plastral lobe of UMNH VP 28352 is more evenly rounded than that of *Denazinemys*, and there is no indentation of the margin at the ends of the femoral-anal sulcus (Fig. 2B). The general shapes and proportions of the plastral lobes are similar between *Scabremys* and *Denazinemys*, but there are distinct differences between the scalation of these areas. Similar to the anterior sulci, the femoral-anal sulcus of UMNH VP 28352 is mostly straight and perpendicular to the midline (Figs. 6A, 6B). This pattern differs from *Denazinemys* specimens recovered in the Menefee Fm. (*e.g.*, WSC 10769), in which those sulci are oriented posterolaterally (Fig. 2B). The anal scales are substantially shorter than wide (*Lucas & Sullivan, 2006*; *Sullivan, Jasinski & Lucas, 2013*). Large portions of the lateral carapace are missing, but sulci partially define the edges of probably the first three inframarginal scales on the right. The lateral portions of the shell are missing posterior to

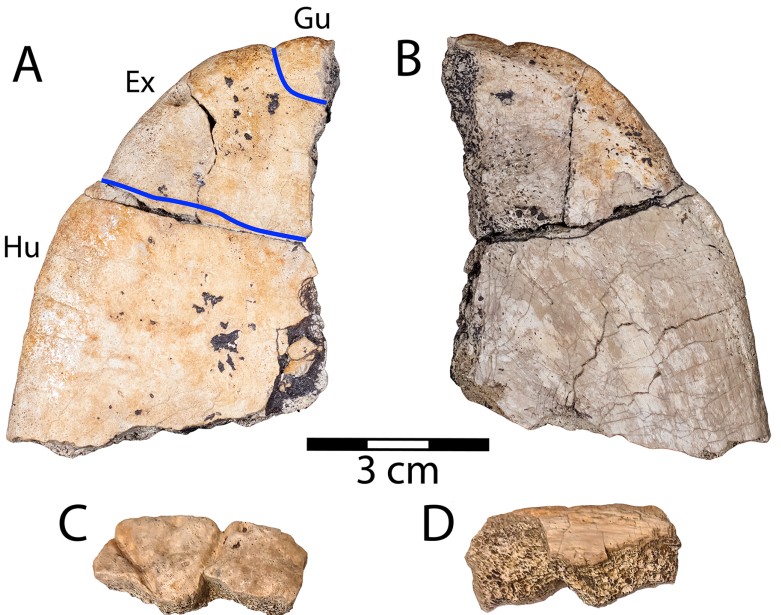

**Figure 7 Right half of the anterior plastral lobe of *Thescelus* cf. *rapiens* (NMMNH P-97384).** (A) Ventral and (B) dorsal views, and an associated partial nuchal showing sulci that define the borders of the first marginal scale in (C) dorsal and (D) ventral views. Abbreviations: Ex = extragular scale; Gu = gular scale; Hu = humeral scale. Blue lines indicate sulci; scale = 3 cm.

approximately peripheral 5 on the left and posterior to peripheral 6 on the right (Figs. 6C, 6D). The straight plastral length of UMNH VP 28352 is 39 cm, and the carapace measures 43 cm along the midline.

*Thescelus Hay, 1908*
*Thescelus* cf. *rapiens Hay, 1908*
Fig. 7

Referred specimen: NMMNH P-97384, right half of anterior plastral lobe and associated partial nuchal.

**Description:** NMMNH P-97384 is an isolated right half of an anterior plastral lobe. It is recognized as a sexually mature baenid based on the fusion of bones (Fig. 7). The ventral surface lacks regular surface texture, but has low, uneven topography and some small, shallow pits (Fig. 7A). This differs from the finely crenulated, pleurosternid-like texture known for *Thescelus insiliens* (*Joyce & Lyson, 2015*). The sulci on the ventral surface are relatively wide with poorly-defined edges, and the lobe margin is indented where it meets them (Fig. 7A). NMMNH P-97384 is recognized as a derived baenid (Baenodda), unlike *N. baueri*, based on substantial midline contact of the gular scales and reduction of the intergular scales (*Brinkman, 2003*; *Joyce & Lyson, 2015*). The presence of gular scales differentiates NMMNH P-97384 from *Scabremys*, and they are more reduced than in *Denazinemys* (*Sullivan, Jasinski & Lucas, 2013*). The anterior plastral lobe is rounded, unlike *N. baueri*, lacks the nodose texture of *Scabremys* and *Denazinemys*, and is smaller

than the other sympatric baenid taxa. On the dorsal side of the lobe, there is a rough patch of bone along the medial edge where the entoplastron is fused with surrounding elements, and there is a beveled free margin that is wider anteriorly and indicates an integumentary transition from keratin to skin. The anterior plastral lobe proportions and sulci of NMMNH P-97384 most resemble *T. rapiens* (PMU.R22) (*Sullivan, Jasinski & Lucas, 2013*: fig. 20.8c-d). *Sullivan, Jasinski & Lucas (2013)* recognized *T. hemispherica Gilmore, 1935* from the Hunter Wash Mbr. of the Kirtland Fm. and noted the difficulty in distinguishing *T. hemispherica* from *T. rapiens*. However, *Joyce & Lyson (2015)* considered the former taxon a *nomen dubium* and retained the latter as valid. Our tentative taxonomic referral to *T.* cf. *rapiens* reflects a degree of uncertainty pending more Menefee material and potential taxonomic clarity within the genus in the future.

NMMNH P-97384 also includes an associated portion of a nuchal that cannot be confidently attributed to a side (Figs. 7C, 7D). It is a marginal component of the carapace with a rounded edge and a superficially damaged ventral side. Two sulci on the dorsal surface diverge toward the margin, and are conspicuously large, measuring approximately two millimeters in depth and width (Fig. 7C). The sulci separate the triangular first marginal scale from the cervical and second marginal scales. This is the only part of the carapace margin of *T. rapiens* (*Sullivan, Jasinski & Lucas, 2013*: fig. 20.8c) where marginal scales are triangular and small.

Helochelydridae *Chkhikvadze, 1970 sensu Joyce et al., 2021*
*Naomichelys Hay, 1908*
*Naomichelys* sp.
Figs. 8A–8D
Referred specimens: WSC 10884, partial nuchal; WSC 10885, hypoplastron fragment.

**Description.** WSC 10884 is a fragment of an element from the periphery of the carapace that is likely anterior due to its obtuse edge. The specimen can be readily identified by the isolated columnar projections that adorn the dorsal surface, which are diagnostic of the broadly distributed North American helochelydrid genus *Naomichelys* (*Joyce, Sterli & Chapman, 2014*; *Joyce, 2017*) (Fig. 8A). A different specimen of *Naomichelys* from the Menefee Formation, NMMNH P-42161, was described by *Lichtig & Lucas (2015)*. WSC 10884 is likely a central portion of the nuchal based on its rounded cross-section, which is thickest in association with a concave margin near the midline of the nuchal or at buttresses of the plastron (*Joyce, Sterli & Chapman, 2014*; *Herzog, 2019*). WSC 10884 is missing sulci or ridges that would occur at buttresses (*Joyce, Sterli & Chapman, 2014*: fig. 8). As with other (especially Late Cretaceous) occurrences (*e.g.*, *Adrian et al., 2019*, *2023*; *Herzog, 2019*), material from this genus is not diagnosable to its type species, *Naomichelys speciosa Hay, 1908*, as it is implausible that one species could achieve such widespread temporal and geographic distribution (see *Joyce, Sterli & Chapman, 2014*).

WSC 10885 is an isolated hypoplastron fragment that likely belonged to the right posterior (inguinal) buttress (see *Joyce, Sterli & Chapman, 2014*: fig. 8.2). The element is irregularly rhomboidal and all edges are broken except the longest, which is relatively thick

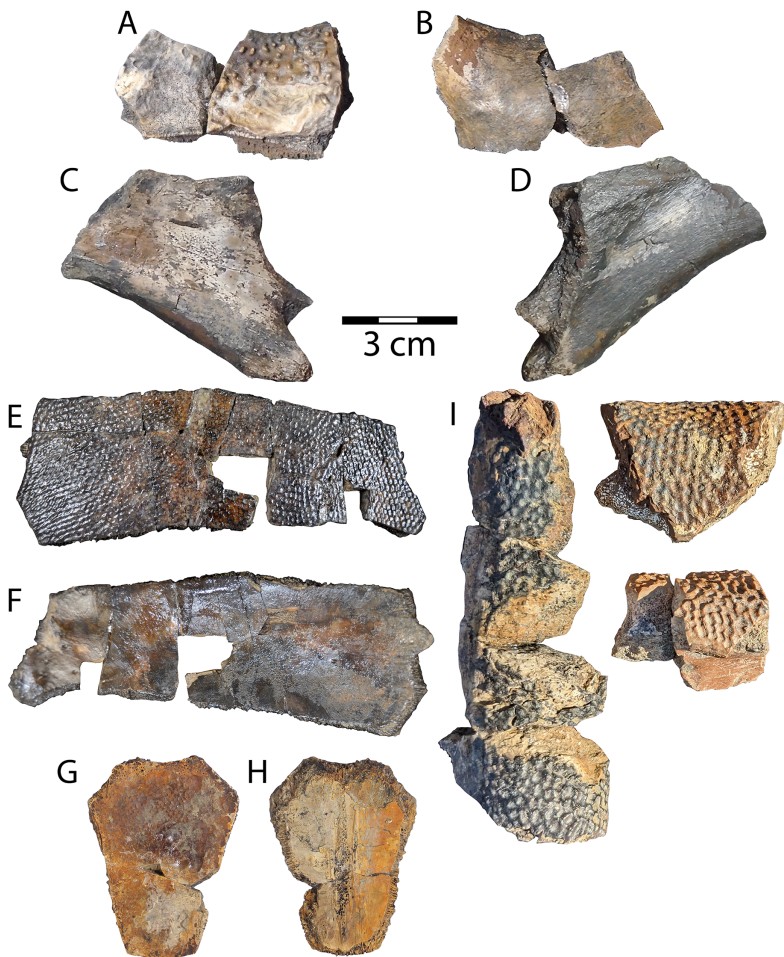

**Figure 8** *Naomichelys* **sp. partial nuchal (WSC 10884).** (A) Dorsal, and (B) ventral views and unassociated hypoplastron fragment (WSC 10885) in (C) ventral, and (D) dorsal views *Adocus* sp. left costal 6 (WSC 10763) in: (E) dorsal, and (F) ventral views; neural cf. 5 (WSC 10765) in: (G) dorsal, and (H) ventral views. (I) *Basilemys* sp. (WSC 10883) indeterminate peripherals. Scale = 3 cm.

and rounded (Figs. 8C, 8D). The thick portion is part of the thickened edge that lines the edge of the buttress opposite the posterior plastral lobe on the dorsal surface. The second longest edge preserves the constricted end of the thick, round edge posteriorly, and portions of costiform processes project from the lateral edge of the element (Figs. 8C–8D). These processes were attachment sites for ligamentous tissue that partially formed the bridge in *Naomichelys*, similar to *Aragochersis lignitesta* (*Pérez-García et al., 2020*; *Lawver & Garner, 2023*).

Testudines *Batsch, 1788*
Pan-Trionychia *Joyce, Parham & Gauthier, 2004 sensu Joyce et al., 2021*
Adocidae *Cope, 1870*
*Adocus Cope, 1868*
*Adocus* sp.

Figs. 8E–8H

Referred specimens: WSC 10763, a sixth left costal; WSC 10765, a (cf. fifth) neural.

**Description.** WSC 10763 is a sixth left costal representing *Adocus* sp. It is referred by diamond-shaped pits arranged in rows, and by the marginal-pleural sulcus occurring significantly medial to the costal-peripheral suture (*Hay, 1908*; *Gilmore, 1919*; *Meylan & Gaffney, 1989*; *Sullivan, Jasinski & Lucas, 2013*). Texture is concentrated at 4–5 pits per mm, similar to *Adocus kirtlandius Gilmore (1919)* from the Kirtland and Fruitland formations (*Sullivan, Jasinski & Lucas, 2013*). The sixth costal likely articulated medially with the sixth and last neural (*Gilmore, 1919*). Sulci on the dorsal surface divide marginal scales 8–9 laterally, pleural scales 3–4 in the middle, and vertebral scale 4 medially (Fig. 8C). The marginal scales are broad, as in the derived Asian congeneric *Adocus kohaku Hirayama et al., 2021*, and unlike most other *Adocus species* (*e.g.*, *Danilov et al., 2013*; *Syromyatnikova, Danilov & Sukhanov, 2013*; *Sonoda et al., 2015*; *Hirayama et al., 2021*). The ventral surface of the costal is mostly flat and smooth, exhibiting minimal relief of the embedded rib; however, the reduced rib head is typical of *Adocus* (*Meylan & Gaffney, 1989*). A small protrusion of the rib end beyond the costal is preserved laterally, however its sutural lateral edge indicates peripheral articulation, differentiating the specimen from Trionychidae (*Vitek & Joyce, 2015*). The maximum width of WSC 10763 is intermediate in size between that of *A. kirtlandius* and *A. bossi Gilmore (1919)* from younger Campanian units in the San Juan Basin (Table 2).

WSC 10765 is an unassociated hexagonal neural that can be attributed to *Adocus* sp. based on a patch of faint surface texture on the anterior end of the dorsal side (Figs. 8E, 8F). Though worn, the pits are consistent in size with better preserved *Adocus* specimens from the Menefee and aligned in regular rows unlike those of trionychids (*Sullivan, Jasinski & Lucas, 2013*; *Vitek & Joyce, 2015*). The neural is broken near a faintly distinguishable sulcus that crosses the dorsal surface on its distal half, just posterior to a transverse crack on the left side (Fig. 8E) (*Meylan & Gaffney, 1989*). The dimensions of the neural and location of the transverse sulcus suggest the specimen is the fifth in the neural series (*Gilmore, 1919*; *Sullivan, Jasinski & Lucas, 2013*: fig. 20.10). The ventral surface has remnants of the bases of vertebral articulations along the midline, with one large midline scar and two thinner flanking ridges (Fig. 8F). The margins of WSC 10765 comprise dense, finely-grooved sutures typical of *Adocus* (*Hay, 1908*; *Gilmore, 1919*). The length of the fifth neural is considerably greater than that of *A. kirtlandius*, but only slightly exceeds that of *A. bossi* (Table 2).

Nanhsiungchelyidae *Yeh, 1966*
*Basilemys Hay, 1902*
*Basilemys* sp.
Fig. 8I
Referred specimen: WSC 10883, peripheral fragments.

**Description.** Remains of *Basilemys* (WSC 10883) were discovered for the first time in the Menefee Fm. in the field season of 2023. The fragments clearly belong to the peripheral

**Table 2 Comparative measurements of Menefee *Adocus* specimens.** Measurements (in cm) from taxa described by *Gilmore (1919)*.

| Taxon | Specimen | Neural 5 length | Costal 6 width |
|---|---|---|---|
| *Adocus* sp. (this study) | WSC 10763 and 10765 | 5.3 | 10.7 |
| *Adocus kirtlandius* | USNM 8593 | 3.4 | 9.3 |
| *Adocus bossi* | USNM 8613 | 5.1 | 14.4 |

ring, where deep pitting occurs on both the dorsal and ventral sides of the broadly rounded peripherals, a diagnostic feature of the genus (*Brinkman & Nicholls, 1993*; *Mallon & Brinkman, 2018*). The bones are clearly distinct from *Adocus* in the size and coarseness of the pits and thicker bone in the peripherals, and cannot belong to a trionychid because they represent peripherals and have a more regular surface sculpture with higher relief. The ornamentation of WSC 10883 is similar in arrangement to that of *Adocus*, but the pits are substantially larger, deeper, and have distinct points that project from the junctions of ridges between pits (Fig. 8I). The ornate surface sculpture of *Basilemys* has also been described as triangular tubercles separated by pits (*Brinkman & Nicholls, 1993*).

Pan-Trionychidae *Joyce, Parham & Gauthier, 2004 sensu Joyce et al., 2021*
Plastomenidae *Hay, 1902 sensu Joyce et al., 2021*
*Helopanoplia Hay, 1908*
*Helopanoplia* sp.
Figs. 9A–9B
Referred specimen: WSC 10767, an edge fragment of a second costal; UMNH VP 36786, an indeterminate costal edge fragment.

**Description.** A pan-trionychid, *Helopanoplia* (*Hay, 1908*), is recognized in the Menefee only from costal fragments and is less abundant than "*Trionyx*" *robustus*. Its pit morphology is distinctly intermediate in size (1.7–4.4 mm diameter) between "*Trionyx*" sp. 'large' and "*Trionyx*" *robustus* (see below). Its recessed areas are less pit-like and flatter than the other Menefee pan-trionychid forms. The ridges between these recessed areas are thin, form polygonal rather than ovoid shapes, and are not rounded. Additionally, the costal free margin is bounded by an upturned ridge approximately 0.75 cm wide on WSC 10767. (Fig. 9A). WSC 10767 preserves the upturned rim of the carapacial "shoulder" that is associated with the second costals in *Helopanoplia*, and UMNH VP 36786 displays the flatter and unornamented lateral margin that is characteristic of some other costals (see *Joyce & Lyson, 2017*: fig. 4A) (Fig. 9B). WSC 10767 and UMNH VP 36786 share similarities with *Helopanoplia* specimens UCMP 194095 and UCMP 194260 from the Kaiparowtis Fm., primarily in the distinctive dorsal carapace sculpture where pits are subequal in size and the borders between them are formed by sharp ridges of variable height, and the morphology of the carapacial rim (*Hutchison, Knell & Brinkman, 2013*: 311; fig. 13.14 a-b; *Joyce & Lyson, 2017*).

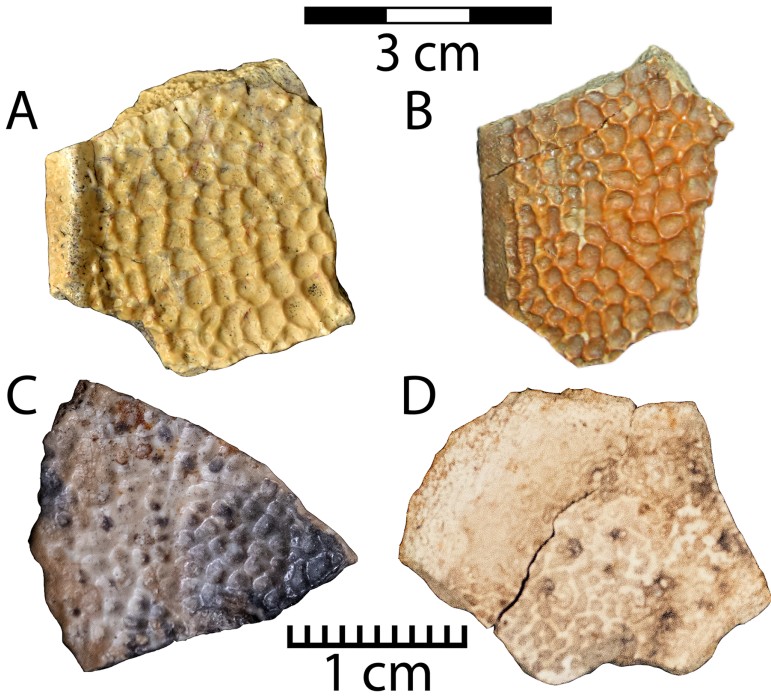

**Figure 9 Material representing two distinct plastomenid forms in the Menefee Fm.** *Helopanoplia* **sp. (WSC 10767).** Costal edge fragment (A), and UMNH VP 36786 (B), a marginal costal fragment; Plastomenidae indet. shell fragments (C) NMMNH P-97383 and (D) NMMNH P-97389. Note different scales between the top (3 cm) and bottom (1 cm) row.

Plastomenidae indet.

Figs. 9C–9D

Referred specimens: NMMNH P-97383 and NMMNH P-97389, shell fragments

**Description.** NMMNH P-97383 and NMMNH P-97389 are small shell fragments of plastomenid turtles, which can be identified based on their characteristic pitted dorsal textures (Figs. 9C, 9D). NMMNH P-97383 is approximately triangular, and one edge is curved while the others are straighter (Fig. 9C). NMMNH P-97389 is irregular in shape and has a more poorly preserved dorsal surface (Fig. 9D). They can be distinguished from the trionychid morphotypes described below by the combination of very small (~1 mm diameter) pits and a very thin (~3 mm) shell. The trionychid form with the smallest pits ("*Trionyx*" *robustus*) differs by having a considerably thicker shell (also recognized by *Joyce, Lyson & Sertich (2018)*). In addition to finely pitted texture, NMMNH P-97383 also has a faint ridge crossing its surface, similar to those of the plastomenid *Gilmoremys gettyspherensis Joyce, Lyson & Sertich (2018)* (Plastomeninae indet. of *Sullivan, Jasinski & Lucas (2013)*). However, the paucity of recovered material for this smallest pan-trionychid in the current Menefee assemblage constrains our referral to a particular plastomenid species (see phylogenetic hypothesis of *Girard et al. (2024)*).

Trionychidae *Bell, 1828 sensu Joyce et al., 2021*
Trionychinae *Lydekker, 1889 sensu Joyce et al., 2021*

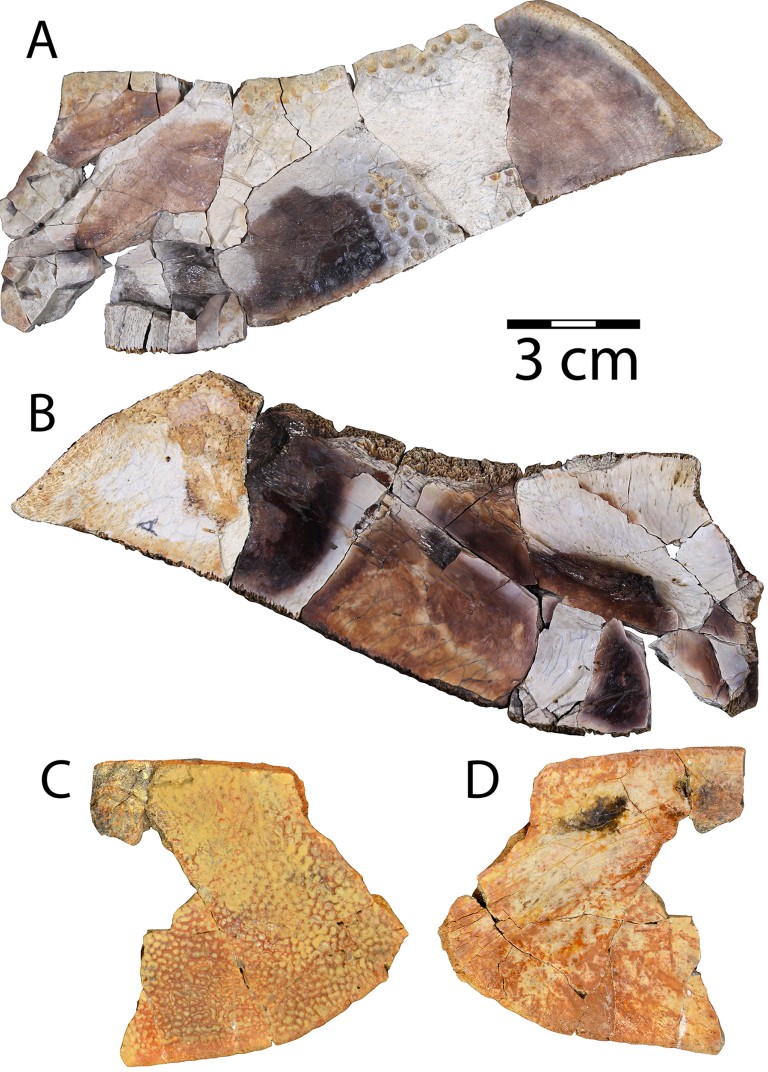

**Figure 10 Material representing two distinct trionychine forms in the Menefee Fm.** Right first costal of *"Trionyx"* sp. 'large' (WSC 10768) in: (A) dorsal, and (B) ventral views; right posterior carapace of *"Trionyx" robustus* (WSC 10766) in: (C) dorsal and (D) ventral views.

*"Trionyx"* sensu *Vitek & Joyce, 2015* (aff. *Aspideretoides sensu Gardner, Russell & Brinkman, 1995*)

*"Trionyx"* sp. 'large'

Figs. 10A, 10B

Referred specimen: WSC 10768, a right first costal.

**Description.** The largest pan-trionychid known from the Menefee is recognized by fragments that often have free costal edges with a border that lacks pitting (Figs. 10A, 10B). Its surface ornamentation is distinct from the smaller forms (see below) and is distinguished by relatively large circular to irregular ovoid pits (2.5–4.5 mm diameter)

(Figs. 10A, 10B). The pits are separated by relatively wide ridges that are low and rounded. WSC 10768 represents a nearly complete first right costal. In most of the superficial layers of pitting on the dorsal side, the most exterior cortical layers are broken away, leaving remnants with variably intact pitted surfaces (Figs. 10A, 10B). Broken edges of fragments from this taxon frequently display the distinctive laminar "plywood-like" internal microstructure of trionychids without the need for magnification (*Scheyer et al., 2007*; *Lichtig & Lucas, 2015*; *Vitek & Joyce, 2015*). Apart from the free margin, sutured edges of the costal are better preserved on the ventral side (Fig. 10B). The shapes of the edges can indicate the shapes of adjacent bones. For instance, the two distinct concavities on the medial side of WSC 10768 suggest two distinct bones (*e.g.*, neurals or preneurals) medial to the first costal, a feature associated with the legacy taxon *Aspideretoides* (Fig. 10B) (*Gardner, Russell & Brinkman, 1995*). The anterior border of the costal suggests that the nuchal was considerably wider than long. The ventral surface also preserves the first rib, which is directed anteriorly toward the nuchal, and its broad, flat head occupies the medial end of the costal (Fig. 10B). WSC 10768 shares similar traits to an unnamed species of *Aspideretoides* ("*Trionyx*" sensu *Vitek & Joyce, 2015*) described from the Kaiparowits Fm. (UCMP 194124), which is referred based on its aforementioned neural area subdivision (*Hutchison, Knell & Brinkman, 2013*: 313; fig. 13.14c). UCMP 194124 is differentiated from *Axestemys splendida Vitek (2012)* by the anterior curve of the first costal as opposed to its straight lateral extension, and WSC 10768 shares this difference (*Hutchison, Knell & Brinkman, 2013*). Dorsal surface sculpture is also similar between UCMP 194124 and WSC 10768 in its even, pitted distribution across the shell, with reduced pit size near the edge of the shell (Fig. 10A). The borders between pits are often wide with rounded or flat tops (*Hutchison, Knell & Brinkman, 2013*: fig. 13.14c).

"*Trionyx*" *robustus* (*Gilmore, 1919*)
Figs. 10C, 10D
Referred specimen: WSC 10766, distal end of right costal 6 or 7.

**Description.** A form attributed to "*Trionyx*" *robustus* has the smaller pits of the two trionychine taxa recognized here from the Menefee Fm. and is best represented by WSC 10766, a portion of the right posterior carapace (Figs. 10C, 10D). Unlike the other forms, the pits are rounded and closely arranged, extending almost completely to the free costal margins. Most pits are similar in size, arranged in rows, and range in diameter from 0.7–1.5 mm. However, they sometimes coalesce and are somewhat amorphous, typically becoming fainter toward the center of the carapace (Figs. 10C, 10D). On its ventral side, a posterior rib diverges slightly posteriorly from the anterior sutural edge and reaches the free margin, where its end is broken and missing (Fig. 10D). WSC 10766 is consistent with a nearly complete carapace of "*T.*" *robustus* from the Fruitland Fm. (PMU.R30; *Sullivan, Jasinski & Lucas, 2013*: fig. 20.15d) in size and morphology. Based on its similarity in size with that specimen, WSC 10766 likely belonged to an individual around 23 cm in carapace length.

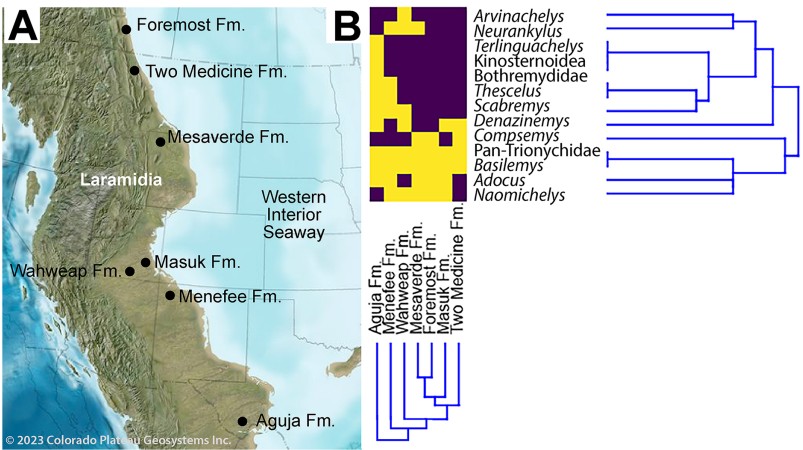

**Figure 11 Paleobiogeographical distribution of turtle taxa along the western shoreline of the Western Interior Seaway during the early-middle Campanian.** (A) Paleogeography of the Western Interior during the middle Campanian; (B) and associated dendrogram of stratigraphic units and taxa. Purple boxes indicate taxa present in the specified formation, while yellow indicates absence. Published sources of turtle assemblages include: Aguja Fm. (*Rowe et al., 1992*; *Lehman & Tomlinson, 2004*; *Sankey, 2005*, *2006*; *Gasaway & Sankey, 2007*; *Lehman & Wick, 2010*; *Lehman et al., 2019*, *2024*; *López-Conde et al., 2019*; *López-Conde, Chavarría-Arellano & Montellano-Ballesteros, 2020*); Foremost Fm. (*Peng, Russell & Brinkman, 2001*; *Cullen et al., 2016*); Masuk Fm. (*Eaton, 1990*); Mesaverde Fm. ( *DeMar & Breithaupt, 2006*, *2008*; *Wu et al., 2023*); Two Medicine Fm. (*Varricchio & Horner, 1993*; *Brandvold, Brandvold & Sweeney, 1996*); Wahweap Fm. (*Holroyd & Hutchison, 2016*). Source map © 2023 Colorado Plateau Geosystems, Inc.               

## Quantitative results

The revised turtle assemblage from the Menefee Fm. is summarized in Fig. 11 and discussed in detail below. Results from the cluster analysis show substantial variation in composition and diversity among the lower-middle Campanian stratigraphic units sampled (Fig. 12). Of these sites, species diversity is highest in the Aguja Fm. and generally higher in southern Laramidian units (Fig. 12B). The most broadly distributed taxa are the helochelydrid *Naomichelys* and pan-trionychian immigrants from Asia (*i.e.*, *Adocus*, *Basilemys*, Pan-Trionychidae spp.). The nanhsiungchelyid *Basilemys* is the only turtle known ubiquitously across the included stratigraphic units. Chelydridae are known from the Mesaverde Fm. northward (*Varricchio & Horner, 1993*; *Brandvold, Brandvold & Sweeney, 1996*; *Peng, Russell & Brinkman, 2001*; *DeMar & Breithaupt, 2006*, *2008*; *Cullen et al., 2016*) (Fig. 12). Kinosternoids (usually cf. *Hoplochelys*) are reported from Campanian strata in Mexico, and the stem kinosternid *Yelmochelys rosarioae Brinkman et al. (2016)* was tentatively identified from the Aguja Fm. and younger strata (*Sankey, Lucas & Sullivan, 2006*; *López-Conde, Chavarría-Arellano & Montellano-Ballesteros, 2020*). Derived baenids are not known north of Utah during the early-middle Campanian. The Wahweap and Menefee formations each had the baenids *Neurankylus* and *Denazinemys* (Fig. 12). *Adocus, Thescelus*, and *Scabremys* are known from the Menefee but have not been reported from the Wahweap, while *Arvinachelys* and *Compsemys* are known from the Wahweap but not the Menefee. The derived baenids *Arvinachelys* and *Scabremys* are also known from higher in the stratigraphic sections of the Kaiparowits Plateau and the San

**Figure 12  Revised bio stratigraphic distribution of turtle taxa in a simplified stratigraphic column of Campanian units in the San Juan Basin.** Blue bars indicate stratigraphic intervals added or extended in the current study to the species inventory provided in *Sullivan, Jasinski & Lucas (2013)*: 360, fig. 20.18). Geological data sourced from *Miller (1984)*, *Miller, Carey & Thompson-Rizer (1991)*, *Lucas et al. (2005)*, *Sullivan & Lucas (2006)*, *Fowler (2017)*, *Pecha et al. (2018)*, and *Ramezani et al. (2022)*. Turtle occurrence data sourced from *Lucas & Sullivan (2006) Sullivan, Jasinski & Lucas (2013)*, *Lichtig & Lucas (2015, 2018)*, and *Joyce, Lyson & Sertich (2018)*.                     

Juan Basin, respectively (*Hutchison, Knell & Brinkman, 2013*; *Sullivan, Jasinski & Lucas, 2013*; *Holroyd & Hutchison, 2016*). Fully marine bothremydids were constrained to the Aguja Fm. in the early-middle Campanian, and the protostegid *Terlinguachelys fischbecki Lehman & Tomlinson (2004)* is known from the same unit.

## DISCUSSION

### Menefee formation turtle assemblage composition

Previous studies reported a turtle assemblage from the Menefee Fm. that was broadly similar (*e.g.*, adocid, trionychid, baenid, and bothremydid taxa) to those of the well-known upper Campanian Fruitland and Kirtland formations, but less diverse and seemingly more primitive (*e.g.*, *Lichtig & Lucas, 2015*). The assemblage we report from the Menefee Fm. confirms the work of previous studies, apart from the ostensible presence of bothremydids, while also increasing the known local diversity of turtle species, primarily by retrograding the stratigraphic ranges of *Neurankylus*, *Scabremys*, *Basilemys*, and *Thescelus* (Fig. 11). This increases the number of baenids, including stem and derived forms, to four and brings the Menefee Fm. more in line with correlative mid-Campanian units elsewhere in Laramidia (Fig. 12). The revised turtle assemblage from the Menefee Fm. exceeds the diversity known from the Ne-nah-ne-zad Mbr. of the Fruitland Fm. due to the presence of

*Naomichelys* (*Sullivan, Jasinski & Lucas, 2013*; *Lichtig & Lucas, 2015*) (Fig. 11). Compared to the Hunter Wash Mbr. of the Kirtland Fm., the Menefee turtle assemblage is missing *Boremys grandis*, *Compsemys*, and a representative of Bothremydidae (*Sullivan, Jasinski & Lucas, 2013*). The ubiquity of *Basilemys* during the middle Campanian in Laramidia contrasts with a patchier distribution for *Adocus*, which has not been reported from the Two Medicine or Wahweap formations (*Varricchio & Horner, 1993*; *Holroyd & Hutchison, 2016*). The persistent endemic helochelydrid *Naomichelys* is represented in New Mexico by only three shell fragments including the two described above; the rarity of the genus in New Mexico suggests it was relatively scarce on the Menefee paleolandscape, or at least in the paleoenvironments represented (*i.e.*, fluviodeltaic and floodplain deposits) (*Lichtig & Lucas, 2015*). *Naomichelys* is known primarily from fragments (*Joyce, 2017*), and it is not known in the San Juan Basin after the Menefee Fm. (*Lichtig & Lucas, 2015*). The Menefee specimens were discovered near the top (WSC 10885) and bottom (WSC 10884) of the stratigraphic range included in our study area, making it likely that *Naomichelys* persisted to the top of the Allison Member, at least locally. Its final occurrences in Laramidia are from the late Campanian of Mexico (*Reynoso, 2006*; *López-Conde et al., 2018*). The surface sculpture and marginal and nuchal morphologies of two of the trionychid forms recognized in the Menefee Fm. strongly resemble those that are characteristic of the genus *Aspideretoides Gardner, Russell & Brinkman, 1995*. For instance, the medial edge of the first costal of "*Trionyx*" sp. 'large' (WSC 10768) suggests the presence of a preneural and a nuchal that is approximately four times wider than long (Figs. 9A, 9B). The presence of a preneural is considered primitive for trionychids (*Gaffney, 1979*; *Meylan, 1984*, *1987*). *Aspideretoides* ("*Trionyx*") is also known from the Campanian of Central Asia (*Vitek & Danilov, 2010*), which has been interpreted as evidence of multiple trionychid dispersal events to North America from Asia. However, problems persist with the diagnostic characters for the genus (*Hutchison, 2000*; *Vitek & Joyce, 2015*). It is plausible that future discoveries will provide enough diagnostic material to increase taxonomic resolution for the trionychid taxa that inhabited southern Laramidia in the middle Cretaceous.

The recognition of *Helopanoplia* in the Menefee Fm. represents an early occurrence of the genus, which is otherwise reported from the Campanian only in the Kaiparowits and Aguja formations, where the deposits are dated to the middle to late Campanian (*Sankey, Lucas & Sullivan, 2006*; *Gasaway & Sankey, 2007*; *Hutchison, Knell & Brinkman, 2013*). This early occurrence suggests a southern Laramidian origin for the genus if the attributions are accurate, and the potential presence of two sympatric plastomenids in the Menefee Fm. matches the diversity of trionychid species. The size distributions of sympatric soft-shelled species also suggest possible niche separation if the current approximate size estimations are correct. Further pan-trionychid discoveries in the Menefee could make this a testable line of inquiry. The pan-trionychids included in the current study are not taxonomically separated for the purposes of the cluster analysis due to taxonomic uncertainty in the group across sampled stratigraphic units.

Increased taxonomic confidence may be possible for *Adocus* in its specific representation through the stratigraphic section, though systematic problems with the

genus remain (see *Danilov et al., 2013*). However, regional comparisons among specimens may elucidate variation in anterior plastral lobe morphologies, which suggests that two species of *Adocus* are present in the Kaiparowits Fm., similar to the upper Campanian portion of the San Juan Basin (*Gilmore, 1919*; *Hutchison, Knell & Brinkman, 2013*; *Sullivan, Jasinski & Lucas, 2013*).

In comparison with Asian *Adocus* species, the costal WSC 10763 exhibits expanded marginal scales similar to *A. kohaku* from the Turonian Tamagawa Fm. of Japan (*Hirayama et al., 2021*) (Fig. 8C). Hirayama and colleagues interpret marginal scale expansion as a derived trait in the *Adocus* lineage. In most Early Cretaceous Asian and North American *Adocus* species, the extension of the marginal scales onto the costals does not start until the fifth marginal, constituting a state which is considered primitive in *Adocus*. In contrast, the expansion of the marginal scales onto the costals begins with the fourth (or occasionally third) marginal in *A. kohaku*, *A. aksary*, *A. amtgai*, and *A. inexpectatus*. Only in *A. kohaku*, *A. amtgai*, and the Menefee *Adocus* sp. are the marginals as wide or wider than the pleurals (see *Hirayama et al., 2021*: table 2). It is noteworthy that the Menefee *Adocus* sp. shares this derived character with the more advanced Asian *Adocus* species, though more research is required to determine the evolutionary significance of intercontinental morphological differences.

## Stratigraphic range extensions

The specimens reported here allow us to confidently extend the stratigraphic ranges of several turtle species to the Menefee Fm. from higher in the San Juan Basin stratigraphic section (Fig. 11). The first is the stem baenid *Neurankylus baueri*, which is known from the younger Kirtland Fm., but not the intervening Fruitland Fm. (*Sullivan, Jasinski & Lucas, 2013*). The same stratigraphic revision is applied to the derived baenids *Scabremys ornata* and *Thescelus*, which are known from the upper Campanian Kirtland Fm. (*Sullivan, Jasinski & Lucas, 2013*). The stratigraphic range of *Basilemys* in the San Juan Basin is also extended back to the Menefee Fm. (Fig. 11). *Basilemys gaffneyi Sullivan, Jasinski & Lucas (2013)* is known from the upper Fruitland and lower Kirtland formations, while the occurrences from the Menefee and Aguja formations predate the earliest named species of *Basilemys* (*B. variolosa Cope, 1876*) from the Judith River Fm. in central Montana (*Langston, 1956*; *Sullivan, Jasinski & Lucas, 2013*; *Mallon & Brinkman, 2018*; *López-Conde, Chavarría-Arellano & Montellano-Ballesteros, 2020*). Consistent with previous studies, the trionychid/plastomenid record in the Menefee Fm. is currently rich in variation and apparently speciose, although most material is fragmentary and isolated, confounding attribution (*Lichtig & Lucas, 2015*). However, we have presented material representing two taxa each belonging to plastomenid and trionychid forms (*Vitek, 2012*; *Vitek & Joyce, 2015*; *Girard et al., 2024*). The consistent recovery of plastomenid turtles from large sandstone channel deposits demonstrates a preference for riverine conditions (*Lyson, Petermann & Miller, 2021*; *Girard et al., 2024*). The diversity of pan-trionychids in the Menefee Fm., along with their size range, suggests that local freshwater hydrological regimes manifested in a variety types, representing both higher and lower energy environments.

## Revised turtle paleobiostratigraphy

Results from the cluster analysis reveal patterns in the diversity and distribution of turtles in Laramidia during the early-middle Campanian, though the sample is not comprehensive and is subject to typical constraints associated with the fossil record (*e.g.*, low sample sizes, taxonomic uncertainty, poor dating; see *Maidment et al., 2021*) (Fig. 12). First, *Naomichelys* has a broad distribution among the sampled sites, and the Menefee Fm. is noteworthy in producing the only three fragments from the taxon that have been reported in New Mexico (*Joyce et al., 2011*; *Lichtig & Lucas, 2015*). Also, the broad pattern of early-middle Campanian baenid distribution is consistent with a low latitude origin for the derived group of baenids (Baenodda), which first appeared in the Cenomanian in Appalachia (*Gehennachelys maini Adrian, Smith & Noto, 2023*) and Turonian of Laramidia (*Edowa zuniensis Adrian et al., 2023*) (*Joyce & Lyson, 2015*). In particular, the stratigraphic extension of *Scabremys* to the Menefee Fm., combined with its presence in the Aguja Fm., demonstrates a broader geographical and temporal range than previously appreciated (*Tomlinson, 1997*; *López-Conde et al., 2019*; *López-Conde, Chavarría-Arellano & Montellano-Ballesteros, 2020*), similar to the spatiotemporal distribution of *Denazinemys* and *Thescelus*. The cluster analysis shows a distinct trend of higher turtle diversity at lower latitudes (Fig. 12B). *Basilemys* and Pan-trionychidae were ubiquitously present in the sampled stratigraphic units, and *Naomichelys* and *Adocus* were the next most widely distributed clades across latitudes. Prior to the late Campanian, derived baenids (*e.g.*, *Arvinachelys*, *Denazinemys*, *Scabremys*, *Thescelus*) only occurred in the Wahweap Fm. and southwards (*Holroyd & Hutchison, 2016*), and kinosternoids were restricted to Gulfian latitudes at the Aguja Fm. (*Brinkman et al., 2016*; *Holroyd & Hutchison, 2016*; *López-Conde, Chavarría-Arellano & Montellano-Ballesteros, 2020*). No sampled taxa were found exclusively in northern latitudes, consistent with previous difficulties in defining a distinctly northern turtle fauna (*Holroyd et al., 2015*).

The revised taxonomic identity of CHCU 81269 from a bothremydid to the baenid *N. baueri* has biogeographical significance. First, it simplifies our understanding of North American bothremydid distribution during the Campanian by constraining it to the Aguja Fm. prior to the late Campanian. It also suggests that there was a hiatus of the pleurodiran clade in the Western Interior between their earliest known invasion into Laurasia during the Cenomanian (*Joyce, Lyson & Kirkland, 2016*; *Pérez-García et al., 2017*; *Adrian et al., 2021*) and *Chedighaii hutchisoni* in the upper Campanian Kirtland Fm. (*Gaffney, Tong & Meylan, 2006*; *Sullivan, Jasinski & Lucas, 2013*). This species is known only from a skull, and questions remain about its phylogenetic relationship with *Bothremys* (see *Gaffney, Tong & Meylan, 2006*) (Fig. 11). The recognition made here of *Neurankylus baueri* in the Menefee Fm. constitutes the earliest occurrence of the species (*Joyce & Lyson, 2015*; *Holroyd & Hutchison, 2016*). This pattern was previously overlooked due to the original misidentification of CHCU 81269 as a bothremydid (*Lichtig & Lucas, 2015*).

Our comparison of *N. baueri* shell dimensions confirms similarity of Menefee specimen WSC 10612 with the younger holotype, and also supports the hypothesis of *Joyce & Lyson (2015)* that anterior lobe dimensions are highly variable in baenids (Fig. 4). Plastral lobe

dimensions have also been used to differentiate between palatobaenines, eubaenines, and *Neurankylus* spp. (*Adrian, Smith & Noto, 2023*).

Numerous previous studies have also reported patterns suggesting a distinct southern Laramidian turtle assemblage (*Tomlinson, 1997*; *Hutchison et al., 1998*; *Rosa & Cevallos-Ferriz, 1998*; *Brinkman, 2003*; *Brinkman & De La Rosa, 2006*; *Sankey, Lucas & Sullivan, 2006*). For baenids in particular, *Joyce & Lyson (2015)* recognized *Boremys pulchra Lambe, 1906*, *Neurankylus eximius Lambe, 1902*, and *Plesiobaena antiqua Lambe, 1902* as northern taxa, and six southern taxa: *Arvinachelys goldeni Lively, 2015*, *Boremys grandis Gilmore, 1935*, *D. nodosa*, *N. baueri*, *S. ornata*, and *T. rapiens*. Though various hypotheses of provinciality and endemism have been vigorously debated, large scale distribution patterns are not broadly consistent in timing or extent across faunal groups, and a clear consensus has not been established regarding the latitudinal ranges of purported provinces within the Western Interior Basin (WIB) (*Lucas et al., 2016*; *Maidment et al., 2021*). However, refined geochronology of Campanian continental strata in the WIB has revealed temporal overlap among the richest dinosaur-bearing intervals in the Dinosaur Park Fm. of Alberta, the Judith River and Two Medicine formations of Montana, and the Kaiparowits Fm. of Utah (*Ramezani et al., 2022*). Given the expanse and complexity of geological and climatic change that is included in the history of the WIB, perhaps a single explanatory model is unreasonable to expect, and may be impossible given the inherent uncertainty and complexity that exists in the fossil record. Rather, constraining the temporal, spatial, and taxonomic parameters may focus on the responses of particular biotas across group-specific spatio-temporal parameters. This approach is particularly relevant for San Juan Basin turtles because of uncertainty surrounding the timing of the Laramide tectonism in north-central New Mexico, which is believed to have started around 75–70 Ma. The lack of precise temporal data makes it challenging to pinpoint when this tectonic event first began (*Thacker et al., 2023*). The turtle distributional patterns and stratigraphic range extensions described here demonstrate that turtle assemblages in the San Juan Basin show broad compositional continuity from the Allison Member of the Menefee Fm. through the Kirtland Fm. The likely presence of multiple trionychid taxa in the Menefee that resemble younger taxa suggests endemism in the group prior to the upper Campanian (Fig. 11). Overall, the revised Menefee assemblage presented here increases the known diversity of turtles in the middle Campanian of the San Juan Basin, aligning it more closely with younger strata, particularly the Fruitland Fm. and Hunter Wash Mbr. of the Kirtland Fm. (Fig. 11) (*Sullivan, Jasinski & Lucas, 2013*). Future collection efforts in the Menefee may further reduce the faunal differences in the sequence, which currently involve the taxa *Boremys grandis*, Kinosternoidea, and any marine taxa (bothremydids or chelonioids).

## CONCLUSIONS

The Menefee Fm. turtle discoveries reported here update our understanding of the San Juan Basin turtle fauna in the temporal interval preceding the rich deposits of the later Campanian. Compared to the last published inventory of the Menefee Fm. turtle assemblage (*Lichtig & Lucas, 2015*), we demonstrate a stratigraphic range extension of

approximately three million years in the San Juan Basin stratigraphic sequence for *Neurankylus baueri*, *Scabremys*, *Thescelus, Basilemys*, and Plastomenidae (Fig. 11). These additions to the Menefee turtle fauna increase its similarity with the younger Fruitland and Kirtland formations and stratigraphically correlative units, especially in southern Laramidia (*e.g.*, Wahweap and Aguja formations) (Fig. 12). Two Menefee trionychids are referred to trionychine taxa that are similar to species that have been assigned to the genus *Aspideretoides* (or "*Trionyx*"), and two more are referred to plastomenid taxa (*Hutchison, Knell & Brinkman, 2013*). The occurrence of one of these, *Helopanoplia*, is early and is similar in age to material from the Kaiparowits and Aguja formations, suggesting a southern origin for the genus. Its presence in the Menefee Fm., along with "*Trionyx*" sp. 'large', suggests a particular similarity with pan-trionychid taxa described from the Kaiparowits Fm. (*Hutchison, Knell & Brinkman, 2013*). The relatively small indeterminate plastomenid could represent *Gilmoremys gettyspherensis* (*Joyce, Lyson & Sertich, 2018*) or a related form, but more material is needed to confirm. *Adocus* specimens from the Menefee are generally larger than *A. kirtlandius* (*Sullivan, Jasinski & Lucas, 2013*), but more material is required to make specific attributions. We describe two fragments of the helochelydrid *Naomichelys*, which constitute the second and third specimens known from the Menefee Fm. and New Mexico more broadly (*Lichtig & Lucas, 2015*). This suggests local and regional rarity of the genus, which is relatively close in time to its final occurrences in Laramidia, in Mexico during the late Campanian (*Reynoso, 2006*; *López-Conde et al., 2018*). The current study increases the diversity of baenids known from the Menefee Fm., including the stem species *N. baueri* and three derived forms that are regionally distributed: *Denazinemys*, *Scabremys*, and *Thescelus* (Fig. 11). We reassign CHCU 81269 to *N. baueri*, and have seen no evidence for the presence of a bothremydid in the Menefee Fm. Though the fossil material described here improves our understanding of the turtles of the Menefee Fm., the unit requires a considerable amount of additional work given its temporal and geographic span, apparent biodiversity, general lack of lateral stratigraphic continuity, and sparse geochronological record (*Lucas et al., 2005*).

## ACKNOWLEDGEMENTS

The authors would like to thank the large number of volunteers who have worked in Menefee Fm. field efforts since 2011, especially those from the Southwest Paleontological Society. Staff and volunteers at the Western Science Center have also contributed substantially to fieldwork and skillful preparation of numerous Menefee fossils. Among these, special appreciation goes to Lab Manager Leya Collins and volunteer Joe Reavis. A great deal of work, expertise, and assistance was also provided by the Zuni Dinosaur Institute for Geosciences. We thank staff at the Natural History Museum of Utah, especially Carrie Levitt-Bussian, for access to collections and arranging loan of specimens to the Western Science Center, and we thank volunteer preparators Paul Boyle and Lewis "Hook" Ershler for their work on UMNH VP 28352. At Chaco Culture National Historical Park, we thank Cynthia Wiley, Claire Kittell, Lori Stephens, and Denise Robertson for

facilitating access to CHCU 81269. We thank staff at the New Mexico Museum of Natural History and Science, especially Anthony Fiorillo, Nicole Volden, and Brian Grace. At the BLM, we thank Philip Gensler and Cody Walton.

## INSTITUTIONAL ABBREVIATIONS

| | |
|---|---|
| **CHCU** | Chaco Culture National Historical Park, Nageezi, New Mexico (collection housed at Hibben Center for Archaeological Research, University of New Mexico, Albuquerque, New Mexico) |
| **NMMNH** | New Mexico Museum of Natural History & Science, Albuquerque, New Mexico |
| **PMU** | Paleontologiska Museet, Uppsala, Sweden |
| **ROM** | Royal Ontario Museum, Toronto, Canada |
| **UCMP** | University of California Museum of Paleontology, Berkeley, California |
| **UMNH** | Natural History Museum of Utah, Salt Lake City, Utah |
| **USNM** | National Museum of Natural History, Smithsonian Institution, Washington, D. C. |
| **WSC** | Western Science Center, Hemet, California |

### Funding

This work was supported by the Western Interior Paleontological Society and David B. Jones Foundation. The funders had no role in study design, data collection and analysis, decision to publish, or preparation of the manuscript.

### Grant Disclosures

The following grant information was disclosed by the authors:
Western Interior Paleontological Society and David B. Jones Foundation.

### Competing Interests

The authors declare that they have no competing interests.

### Author Contributions

- Brent Adrian conceived and designed the experiments, performed the experiments, analyzed the data, prepared figures and/or tables, authored or reviewed drafts of the article, and approved the final draft.
- Heather F. Smith conceived and designed the experiments, performed the experiments, analyzed the data, authored or reviewed drafts of the article, and approved the final draft.
- Andrew T. McDonald conceived and designed the experiments, authored or reviewed drafts of the article, curation, preparation, project supervision, and approved the final draft.

## Field Study Permissions

The following information was supplied relating to field study approvals (*i.e.*, approving body and any reference numbers):

Fieldwork was approved by the Bureau of Land Management (permits NM11-005S, NM12-03S, NM16-11S, NM18-03S, and NM24-04S), and permission to study the specimens was granted by Western Science Center and Natural History Museum of Utah in Salt Lake City, Utah; New Mexico Museum of Natural History & Science in Albuquerque, New Mexico; and National Park Service (permit CHCU-2023-SCI-0008).

## Data Availability

The data are in the body of the text, and there is no other code.

## Supplemental Information

Supplemental information for this article can be found online at http://dx.doi.org/10.7717/peerj.19340#supplemental-information.

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
