# Peer review of "A revised turtle assemblage from the Upper Cretaceous Menefee Formation (New Mexico, North America) with evolutionary and paleobiostratigraphic implications"

_PeerJ, doi:10.7717/peerj.19340_

## Round 0.1 · original submission · Minor Revisions

This study described new fossil materials and provide insight into turtle evolution during the Campanian and Cretaceous turtle paleobiostratigraphy. After some minor revisions mainly about the systematics and materials and methods parts, the paper should be ready to be published.

Reviewer 1 ·

Basic reporting

This ms. seems important for understanding the detailed stratigraphic distributions of the Late Cretaceous turtles in North America, although more complete materials should be desirable.

Experimental design

Systematics of the genus Adocus should be discussed in more detail, as a costal shows a very broad extent of marginal scale; very advanced feature as in some Asian taxa such as A. Kohaku (see Hirayama et al., 2021).

Validity of the findings

These materials are very significant as stratigraphic data by not only by its taxonomic diversity, but also by its very exact geological record.

Additional comments

Systematics of the genus Adocus should be discussed in more detail, as a costal shows a very broad extent of marginal scale (Fig. 8E); very advanced feature as in some Asian taxa.

Annotated reviews are not available for download in order to protect the identity of reviewers who chose to remain anonymous.

·

Basic reporting

Overall the authors do a good job at introducing and describing various fossils from this particular stratum, and then putting them into a larger context.

Their language usage is largely clear throughout, although there are a few instances where more clarity is needed, and I have marked these instances in the attached annotated PDF.

The references are all fairly good, there may be some others that could be utilized on the relevant taxa, but they are not necessarily needed, they would just provide a more thorough list.

The figures in general are all fairly well done, although there is some edits needed with some of the captions explaining them, including making sure that some of the elements of some figures are clarified.

Experimental design

The major aspects of the paper deal with describing material, doing some relatively simple analyses, and putting things into context. The analyses are all explained clearly. However, while it is not directly part of the analysis, part of the methods include the description of material. While the material is described clearly, it seems there is other material from the formation that was not assignable to a more accurate taxonomic identification. While maybe not necessary, it would be nice to know if there was other material assignable only to higher taxonomic levels, such as was there material assignable to Baenidae indet.? Or Pan-Trionychidae indet.? Knowing that would make things more thorough, even if that information was just available as part of a table.

Validity of the findings

The descriptions and identifications are all well done, although they could use a bit more comparison with some known taxa and specimens.

Regardless, the data presented, and the analyses, do support the hypotheses put forward, so the findings of the study, as they are presented, are valid.

Additional comments

I have made a number of comments throughout the paper as part of the attached annotated PDF. Much of what I have should be fairly simple things, often dealing with some clarifications on some aspects, or on asking for some more comparisons for the described material.

Providing a table with all the material and its IDs would be useful, and could be a way to include other Menefee material without having to describe everything, especially if they are only identifiable to higher taxonomic levels.

One other thing to mention is that there are numerous places throughout the text where statements are made such "the specimen described previously", or "the specimen mentioned by authors X and Y". In these instances, I highly suggest providing the specimen numbers when they are mentioned. It ensures there is no confusion as to what specimen is being discussed, whether its earlier in this paper or within another, and can help when the next author or authors either looks back over the Menefee to update and revise it, or uses it to compare with material from other strata.

Overall this is fairly well done, and a bit more work to provide a few more comparisons and clarify some points will make this a nice contribution to the literature. I look forward to seeing it published.

·

Basic reporting

The language used throughout the manuscript is accurate and concise. The references all appear to be accurate with one exception. According to Joyce et al., 2021, the taxonomic authority for Baenidae is Cope, 1873 (see reference below), not Cope, 1882.

Cope ED, On the extinct vertebrata of the Eocene of Wyoming, with notes on the geology. Annual Report of the United States Geological Survey of the Territories, 6, 545–649.

Figures:

1. The figures are excellent, especially those that include line drawings. The artist(s) of these illustrations should be commended for their work! In the review pdf, the captions for figures 11 and 12 appear to be reversed. The image shown for figure 11 seems to correspond with the caption for figure 12.

2. Would it be possible to include a brief explanation of blue vs yellow color scheme in the two-way hierarchical clustering shown in Figure 12? I assume that yellow indicates “presence” vs blue “absence” but I could also be misreading.

3. I am confused as to why there are cladograms associated with the formations and taxa shown in Figure 12B. Is the taxa cladogram part of a phylogenetic analysis conducted by the authors? Were these trees taken from another study? What does the cladogram shown for the various stratigraphic units included in the present study indicate about the relationships between these formations?

Experimental design

This article meets all PeerJ standards for experimental design.

Validity of the findings

No comment

Additional comments

Adrian et al. do an excellent job of providing a thorough and detailed update to the turtle fauna of the Menefee Formation. They examine turtle material collected from this formation during the last 10 years and use that material to revisit the evolution and paleobiostratigraphy of Late Cretaceous turtles in Laramidia. The updated Menefee Fm. turtle fauna described by the authors provides considerable stratigraphic range expansions for several taxa within the San Juan Basin and contributes to our understanding of turtle distribution in southern Laramidia during the mid-Campanian. I can find no significant problems with the accuracy of the information in this study, the analyses performed by the authors, or the conclusions they draw. Overall, I thoroughly enjoyed this manuscript, and I think this is a very informative study. The authors do an excellent job of succinctly summarizing data taken from a large number of specimens that were collected over many years. Faunal updates are an important and often overlooked type of paleontology paper and I can only recommend that this article be published in PeerJ.

---

## Round 0.2 · accepted · Accept

All of the reviewers' comments have been addressed.

The Section Editor noted that Figure 4 has a poor resolution - this should be improved for the final publication (e.g., during proofing phase).